# Exploring Routes to Coexistence: Developing and Testing a Human–Elephant Conflict-Management Framework for African Elephant-Range Countries

Eva M. Gross [1,*], Joana G. Pereira [2], Tadeyo Shaba [3], Samuel Bilério [4], Brighton Kumchedwa [5] and Stephanie Lienenlüke [6]

1    Linking Conservation and Development, 69198 Schriesheim, Germany
2    cE3c-Centre for Ecology, Evolution and Environmental Changes, Faculdade de Ciências, Universidade de Lisboa, Campo Grande, 1749-016 Lisboa, Portugal; jgopereira@fc.ul.pt
3    Royal Place Limited, Lilongwe P.O. Box 30131, Malawi; tadeyo@royalplaceltd.com
4    Wildlife Conservation Society, Niassa Special Reserve, Maputo P.O. Box 163, Mozambique; bilerio4@hotmail.com
5    Department of National Parks and Wildlife, Lilongwe P.O. Box 30131, Malawi; bright.kumchedwa@gmail.com
6    Deutsche Gesellschaft für Internationale Zusammenarbeit, Global Project Partnership against Wildlife Crime in Africa and Asia, 65760 Eschborn, Germany; stephanie.lienenlueke@giz.de
*    Correspondence: mail@eva-gross.com

**Abstract:** Creating a future for elephants and people is a highly complex and dynamic challenge, involving social, behavioral, and ecological dimensions as well as multiple actors with various interests. To foster learning from human–elephant conflict (HEC) management projects and share best practices, a study was conducted to review the management of conflicts between elephants and humans in 12 African countries by qualitative expert interviews. Based on this information, a HEC management framework was developed in a two-tiered process. In the first phase, the theory of the framework was developed. In a second phase, the theoretical framework was validated and adjusted through stakeholder participation in two southern African projects (in Mozambique and Malawi). This holistic approach considers environmental as well as social, political, cultural, and economic factors directly or indirectly affecting interactions between people and wildlife. The framework integrates six interlinked strategies to guide managers and conservation practitioners to address HWC drivers and mitigate their impact. A legal environment and spatial planning form the basis of the framework. Social strategies, including meaningful stakeholder engagement and design of appropriate institutional structures and processes are considered the heart of the framework. Technical and financial strategies represent its arms and hands. At the top, monitoring steers all processes, provides feedback for adjustment, and informs decisions. The integration and coordination of these six strategies has great potential as a guiding route to human–wildlife coexistence in Africa and elsewhere.

**Keywords:** human–wildlife conflict; community-based conservation; participatory process; institutional fit; *Loxodonta africana*

## 1. Introduction

Many of the 37 African elephant-range countries report an increase in human–elephant conflicts (HEC) [1]. While elephants and humans have evolved in Africa, having a 250,000-year history of cohabitation, they continuously competed for resources to some extent, wherever they shared a landscape [2–4]. HEC today is driven by multiple factors [5–8]. Depleted, destroyed, and fragmented habitats force elephants to shift into other areas, changing their traditional movement patterns and spatially isolating populations [9,10]. Elephant habitat loss is mainly caused by human population growth and land-use transformation, which significantly changed the dynamics of social and ecological systems [11].

These drivers, coupled with climate change, increase competition for scarce water and food sources during dry periods, which pull elephants into human habitations attracted by highly palatable crops and permanent water reservoirs [12–16]. Furthermore, with the end of the massive ivory crisis in the late 1990s and the end of decades of civil wars in various elephant-range states, elephant populations were slowly recovering in some areas and moving back into areas they had once populated [17–21]. Many of these areas were now populated by people, who did not know how to coexist with elephants (anymore), resulting in a drastic rise in HEC [22]. In fact, 70% of the elephants' range is currently unprotected, and it is important to consider that the majority of the land that African elephants (*Loxodonta africana*) are utilizing is communal and private land without a conservation status [1,23]. This not only brings elephants into close contact with local communities, easily sparking HEC, it also evidences the importance of focusing conservation efforts on multi-use landscapes, notably agricultural areas [24]. When elephants damage crops and human properties or lives, losses can be substantial [25]. If not appropriately responded to, this can easily influence the attitude towards the species and conservation issues in a negative way [26,27]. On the other hand, elephants can suffer from retaliatory killings or being harmed during human activities to defend habitations or fields [28,29]. In addition to these direct losses, intangible costs of living with elephants and psychological stress make it difficult for people to develop tolerance to elephants living within their community [30–32].

Without doubt, the increase of HEC is ranked as one of the most serious threats to the species' survival [13]. For this reason, the management of HEC has gained high attention over the past decades [11,33–35]. Situations in which people are negatively impacted physically (e.g., injuries), economically (e.g., damage of crops and property), or socially and psychologically (e.g., freedom, security, identity, dignity) by wildlife, and in which humans threaten wildlife populations and the environment, are considered as HWC [36–38]. In most cases, such conflicts have underlying causes that emerge or are intensified by disagreements between groups of people [39], which explains that a dispute between people and wildlife may represent only a surface manifestation of a much more deep-rooted social conflict between people [40,41].

As HEC is actually a conflict between people about the management of elephants and the use resources, the political economy and social dimensions need to be taken more into the focus of HEC analysis and management to move towards coexistence between elephants and people [42,43]. The great challenge conservation institutions face when targeting HEC management is the high complexity and dynamics within the socio-ecologic systems involved [44–47]. This complexity calls for a transdisciplinary approach, which may exceed single institutions' capacities [48–50]. To fully understand the conflict drivers, their impacts, and interconnectivity, holistic views involving multiple factors (e.g., ecologic, social, cultural economic factors) need to be employed [40,51–53]. Adding to this, HWC does not only touch upon conservation concerns, but also social and development issues and thus calls for transdisciplinary analysis and cross-sectoral integration of planning and action (e.g., development, health, conservation, security, waste management) [11,54–56].

The broad and complex subject of HEC has been studied by research and conservation institutions since the 1990s [35]. At first, studies were particularly focusing on technical measures through prevention (e.g., use of fences and deterrents) and mitigation (e.g., by translocations, selective culling, monetary compensation) for conflict management and on monitoring [5,11,57,58]. Over the past ten years, the discussion about HEC and HWC in general have been broadened. Social aspects and the human dimension became more relevant for research, focusing, e.g., on understanding the relationships between people in conflict over wildlife [59], studying how tolerance is shaped [60], which role risk perception plays in the conflict [61–63], and on the importance of social values to conservation [64].

The scientific debate around HEC/HWC and coexistence has shown that these powerful terms may have varying definitions and evoke various reactions and emotions [45,65]. Whether we define the journey from conflict to coexistence as a continuum [66] or recognize lack of clear delineation [54] or the need to introduce a new term and concept [67], the call

to change the state of conflict to something more beneficial and more equitable for those living with it is generally supported in theory [56,68].

As this thinking takes hold, implementers face the reality of short project timelines, insufficient resources, and political pressure. Despite well-intended ambitions, HEC management is too often driven by the desperate search for quick solutions. Conservation institutions implement projects with mainly technical focuses (e.g., fences, drones or geofences) without properly involving the local communities [69–71], lacking sustainability and having disconnected land-use planning [22]. Such quick solutions are detached from the complexity and dynamics of conflicts and fall short on mechanisms for systemic change. Importantly, they miss out on taking those who live with wildlife on board and do not reconcile relational and structural conflicts, resulting in short-term outcomes [47].

Recent holistic and integrated HWC management approaches were developed to tackle some of these gaps. Widening the angle from one species to multiple species, which may be interrelated, further fostered system thinking [72]. An integrated approach to managing HWC was introduced by WWF, called the Safe systems approach [33,73], highlighting the need to integrate various management measures to de-couple the damage and threat from a rise in wildlife populations. Additionally, Marchini et al. [54] developed a conceptual framework to guide conservation projects to set clear goals and take actions based on evidence and structured, participatory decision making. Furthermore, the Human–Wildlife Conflict and Coexistence Specialist Group of the International Union for Conservation of Nature (IUCN) published position statements and briefing papers in preparation of guidelines that emphasizes the uniqueness of every HWC case due to varying ecological, social, and economic factors and acknowledges the need to carefully design processes and foster capacity building and learning exchange [39].

While the uniqueness of every HWC case is fully acknowledged, practitioners from conservation institutions feel a daily and pressing need to adequately deal with HEC. Numerous measures for HEC management have been trialed and tested and controversially discussed in science and practice within the past decades. These measures involve various management areas, such as spatial planning, policies, technical strategies, social strategies, and monitoring. However, because they are conducted by different people or even different institutions, they are lacking integration and communication [74]. As knowledge and experience have grown, we believe it is time to bring lessons learned from science and different management areas together and define some guidance for those involved in handling the delicate interplay of people and elephants. While it is not possible to advise on a standard solution, in this paper, we draft a framework that considers HEC in its entirety and complexity and proposes a way forward in exploring practical solutions aimed towards coexistence. The framework creates an overall understanding of requirements for integrated HEC management and points at potential gaps, which need to be addressed. The framework offers scientists and practitioners an overview of which strategies and processes need to be concurrently or stepwise integrated to develop a sustainable and impactful change towards the coexistence of people and elephants.

## 2. Materials and Methods

This HEC-management framework was developed in a two-tiered process (Figure 1). In the first phase, the theory of the framework was developed based on scientific and grey literature on HEC management and qualitative expert interviews. The outcome of phase 1 was a draft theoretical framework, which, in the second phase, was validated and adjusted through stakeholder participation in two southern African projects.

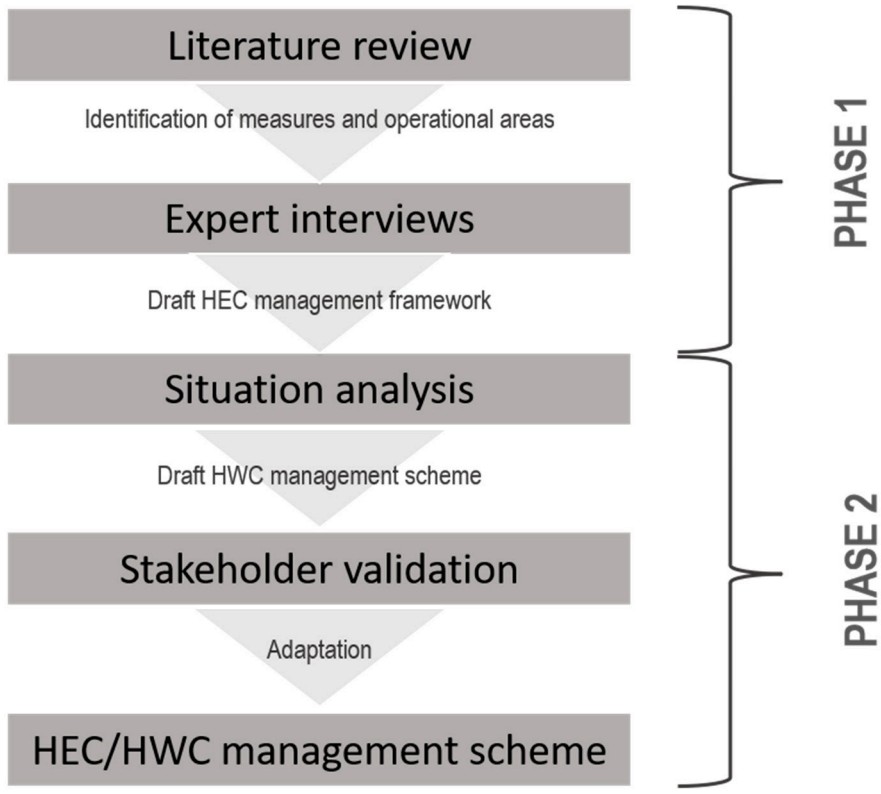

**Figure 1.** Overview of phase 1 and phase 2 of the study process for the development of an integrated and holistic HEC/HWC management framework.

*2.1. Phase 1: Building the Theoretical Framework*

Within the German Corporation for International Cooperation GmbH (GIZ) global project "Partnership against Poaching and Illegal Wildlife Trade", commissioned by the German Federal Ministry for Economic Cooperation and Development (BMZ) and the Federal Ministry for the Environment, Nature Conservation and Nuclear Safety, and Consumer Protection (BMUV), a study was conducted in 2019 to review the management of conflicts between elephants and people in Africa [69]. Therefore, tools and measures used in HEC management were grouped into six overarching strategies (monitoring, spatial planning, policies and legislation, technical strategies, social strategies, and financial strategies) and their impacts and limitations (pros and cons) were summarized. As research articles and reports may not always reflect the views of conservation managers, the study was supplemented by qualitative expert interviews [75]. Therefore, elephant conservation projects implementing HEC management in Savannah habitats from southern, eastern, and western African countries were selected, whereby a focus was set on areas with important elephant populations and high HEC rates (Figure 2). Projects from 13 African countries were selected (Benin, Botswana, Burkina Faso, Ghana, Kenya, Malawi, Mozambique, Namibia, Nigeria, Tanzania, Uganda, Zambia, and Zimbabwe), which identified their most experienced HEC managers for interviews. The interview partners were 23 experts from Anglophone and Francophone African countries, who had spent considerable time working for local or international non-governmental or governmental conservation agencies/organizations on the topic of HEC (Table 1). The interviews were semi-structured (see Supplementary Materials) with open-ended questions, targeting narrative in-depth answers, as used in qualitative social research [76]. The interview was constructed based on Helfferich [77] and organized into three blocs of questions: (A) considered the character of interactions between people and elephants, the value of elephants for people and relationships between various interest groups; (B) explored the various HEC management measures used, how they were implemented and maintained, and which effects, successes, challenges, and limitations

were observed; and (C) included lessons learned from HEC management experience of the past, conditions that need to be fulfilled to maintain healthy elephant populations within the next 30 years and requirements of the institution to create coexistence of people and elephants in the future. The interviews took 50 to 180 min and were carried out online as single-expert interviews and were recorded and hand transcribed for analysis [78]. Participants were informed about the purpose of the research in a signed consent form and agreed on the method of data collection, storage, and analysis and the anonymized publication of collected data and confirmed their voluntary participation in the study.

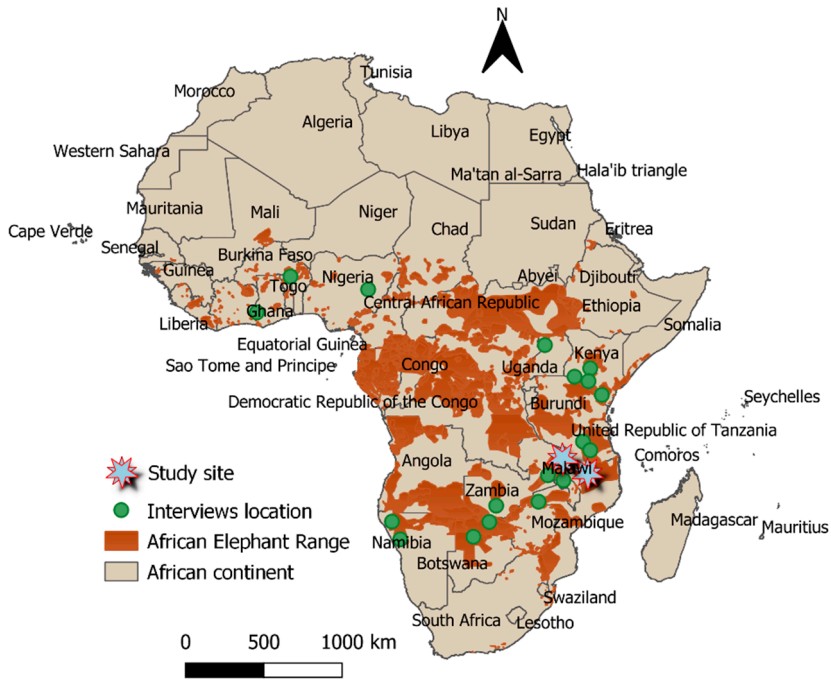

**Figure 2.** African Elephant Range 2015, which includes known, possible, and doubtful ranges [13], locations of expert interviewees of this study—phase 1 (green dots), and study areas for adaptation and validation of experts interviewees—phase 2 (stars markers).

**Table 1.** Qualitative information sources in phase 1 (framework development), expert interviews.

| Phase 1: Expert Interviews | |
| --- | --- |
| **Country** | **Institution** |
| Benin | GIZ/RBT-WAP |
| Botswana | EcoExist |
| Burkina Faso | World Wide Fund for Nature |
| Ghana | University of Science and Technology |
| Kenya | African Wildlife Foundation |
| Kenya | Save the Elephants |
| Kenya | Mara Elephant Project |
| Kenya | Save the elephants |
| Kenya | World Wide Fund for Nature, Tanzania |
| Malawi | Lilongwe Wildlife Trust |
| Mozambique | Niassa Carnivore Project/TRT Conservation Foundation |
| Mozambique | Niassa National Reserve/Wildlife Conservation Society |
| Namibia | Elephant-Human Relation Aid |
| Namibia | Integrated Rural Development and Nature Conservation |

**Table 1.** *Cont.*

| Nigeria | Wildlife Conservation Society | | |
|---|---|---|---|
| Tanzania | Pams Foundation | | |
| Tanzania | Southern Tanzania Elephant Program | | |
| Uganda | Kibale Forest School Program | | |
| Zambia | Conservation South Luangwa | | |
| Zambia | Elephant Connection Research Project | | |
| Zambia | Frankfurt Zoological Society | | |
| Zimbabwe | Connected Conservation | | |
| **Phase 2: Adaptation and validation** | | | |
| **Country (PA)** | **Methodology** | **No. of units** | **No. of participants** |
| Malawi (VMWR&NNP) | Focus group discussions | 9 | 92 |
| | Household surveys | 85 | 85 |
| | Expert and key-informant interviews | 38 | 51 |
| | Extension worker surveys | 79 | 79 |
| | Validation meetings and workshops | 10 | 111 |
| | *Total Malawi (VMWRandNNP)* | | *418* |
| Mozambique (NSR) | Focus group discussions: | 7 | 80 |
| | Household surveys | 196 | 196 |
| | Expert and key-informant interviews | 15 | 20 |
| | Validation meetings and workshops | 2 | 19 |
| | *Total Mozambique (NSR)* | | *315* |

### 2.2. Phase 2: Stakeholder Validation and Adaptation

From January 2020 to December 2021, the applicability of the holistic and integrated HEC management framework was tested in two large-scale conservation projects, Niassa Special Reserve (NSR), Mozambique, and Vwaza Marsh Wildlife Reserve and southern Nyika National Park (VMWR and NNP), Malawi (Figure 1). Due to the multi-species approach in these areas, the framework was adapted to a setting with multiple herbivorous and carnivorous species. In both cases, local community outreach experts conducted mixed-methods approaches, involving various stakeholders [79,80]. Qualitative data on the impact of HEC/HWC and requirements, needs and challenges in HEC/HWC management were obtained from primary sources through focus group discussions with participatory mapping exercises, household surveys, extension worker surveys, and expert/key-informant interviews (Table 1). Finally, validation meetings and workshops with key stakeholders were carried out. All participants were informed about the purpose of the study and about their voluntary participation and the anonymized publication of collected data.

### 2.2.1. Study Area 1: Niassa Special Reserve (NSR)

NSR in Northern Mozambique is a peopled conservation area with an estimated 58,000 inhabitants [81] living in 44 villages, with a strong interface of people and wildlife. Villages are concentrated in three main areas plus some dispersed enclave villages throughout the reserve. With an area of 42,300 km$^2$, the reserve comprises 31% of Mozambique's

protected land and harbors highly significant populations of wildlife, including the largest populations of circa 3600 elephants, lions (*Panthera leo*) (1000–1200), leopards (*P. pardus*), wild dogs (*Lycaon pictus*) (400–450), sable (*Hippotragus niger*), kudu (*Tragelaphus strepsiceros*), wildebeest (*Connochaetes taurinus*), and zebras (*Equus quagga*) [82,83]. It is connected to the Selous Game Reserve (55,000 km$^2$) in southern Tanzania by the Selous-Niassa corridor and remains connected by a natural corridor of forestry concessions to the Quirimbas National Park (7506 km$^2$) to its east, on the coast of northern Mozambique. This remains one of Africa's largest contiguous wilderness areas [84]. The reserve falls across two provinces (Niassa and Cabo Delgado) and comprises eight districts out of which two district administrative centers are located entirely within its boundaries (Mavago and Mecula). NSR is divided into 18 blocs/concessions, which use the natural resources for generating income through hunting, tourism, and philanthropy.

The local communities within NSR mainly make their living from rain-fed subsistence agriculture, mainly maize, sorghum, and beans. Post-harvest techniques to store and process surplus production as well as skills to develop and access markets for products are low [85]. Additionally, households rely on the consumptive use and trade of the available natural resources, particularly by mining, fishing, honey gathering, and bushmeat consumption and trade [86].

### 2.2.2. Study Area 2: Vwaza Marsh Wildlife Reserve (VMWR) and Nyika National Park (NNP)

VMWR (986 km$^2$) and NNP (3134 km$^2$) are located in northern Malawi and are part of the Malawi–Zambia Transfrontier Conservation Area (TFCA) (32,278 km$^2$), which was formally established in 2015 [87]. The objective of the TFCA is to support the improvement of ecosystem connectivity, efficient and sustainable use, and the management of shared natural resources for biodiversity conservation and socio-economic development in the TFCA.

VMWR is a flat terrain located in the Central African Plateau on the watershed between Lake Malawi and the eastern lip of the Luangwa rift to the southeast of the Nyika Plateau. VMWR is home to many species of ungulates (impala (*Aepyceros melampus*), reedbuck (*Antilope redunca*), kudu, bushbuck (*T. scriptus*), and buffalos (*Syncerus caffer*)) and carnivores (spotted hyaena (*Crocuta crocuta*), leopards, and side-striped jackals (*Lupulella adusta*)), lions are seen occasionally [88]. There are an estimated 300 elephants in VMWR which are heavily dependent on a few water resources remaining available in the dry season (May–November) located in the southern part of the reserve (e.g., Lake Kazuni and South Rukuru River) [88]. VMWR has been a partially fenced reserve over the past two decades. In 2007 the 3200 ha Bambanda Zaro wildlife sanctuary was established on the boundary between Vwaza Marsh and the Lundazi Forest Reserve.

NNP encompasses the Nyika massif, the largest montane complex in south central Africa [89] with an extensive high altitude plateau, which lies between 1800 m and 2606 m, while steep escarpments ring the plateau [90]. Carnivore species include leopards, lions, spotted hyena, and side-striped jackals, ungulates encompass reedbuck, bushbuck, eland, roan antelope, and klipspringer [91]. Due to heavy poaching, fewer than 50 elephants remained in the park, and the population was enforced by 34 elephants translocated from Liwonde NP in 2017, which at the time of conducting this study remained in a 7000 ha sanctuary, pending release once a game-proof fence has been installed on the southern boundary of NNP.

Declared in 1965, NNP was enlarged in 1978 to its present size, which involved resettling about 5000 people. The evictions took place between 1975 and 1978 [92]. The resettlement of these communities has resulted in persistent conflict, which contributes to illegal activities inside the park. Encroachment by the communities living along the borders and constant HWC are ongoing challenges.

The human population within the study area adjacent to VMWR and the southwestern NNP is estimated to be 70,000 [79]. The primary livelihoods of households in

areas surrounding both the VMWR and the NNP are dependent on smallholder rain-fed crops, with some characteristics of forest-dependent communities. Typically, these households have limited livelihood sources outside agriculture production. The majority of the households rely on fuel wood as their main energy source, increasing the pressure on protected area resources. Communities adjacent to VMWR and the NNP are in the lowest income bracket and strongly affected by food shortages [93].

## 3. Results

### 3.1. Six Areas of HEC Management Operation

A number of measures and tools used to prevent negative interaction before it occurs were identified and collected in the literature review process of phase 1. These measures are aimed at deterring elephants from agricultural land and habitations or mitigating damage in case previous actions failed, along with methods to monitor the extent and magnitude of HEC and its management, as well as communication and outreach. Such tools and measures were grouped into six areas of management operation, which are all integral parts in HEC management: political, social, financial, technical, spatial, and monitoring (Table 2). During the expert interviews, these six areas were specifically examined, and insights about requirements and best practices of specific methods and activities were collected. The specific focus was set on understanding the effects, challenges, and limitations of methods used as well as the interplay between them and further requirements for the coexistence of people and elephants in the respective areas. Methods that were identified as effective and valuable by the experts were clustered into 32 tools (Table 2). Furthermore, the interviews validated the six areas, highlighted their interconnectivity, and confirmed that creating long-term perspectives in HEC management involves consideration of all six areas simultaneously. Further lessons learned from the expert interviews is that HEC is seen as a symptom, not a cause (interviewees 00P and 00U). Habitat loss and the arising competition for land and resources as well as other economic, political, and social factors are seen as causes and drivers of HEC. In general, it is understood that the problem of HEC cannot be wiped out completely (interviewee 00I and 00T). There will always be some risk of crop and property damage and negative perceptions by individuals. However, the risk of damage must be reduced to a tolerable level (interviewee 00H).

HEC is a highly complex phenomenon with many levels involved (political, community, social, family, financial, tradition, culture, ecology) (interviewee 00Q) and is strongly dependent on the context (interviewee 00L). Simple solutions, therefore, are not to be expected, and learning about HEC needs to continue (interviewee 00L). Due to its complexity, holistic approaches, which understand and tackle the problem from all sides, are needed (interviewee 00Q), and strong community participation is generally seen as the only way forward (interviewees 00A, 00B, 00N, 00O, 00Q, and 00T). The involvement of local communities in HEC management in a strong participatory way seems crucial for achieving a peaceful coexistence of people and elephants (interviewee 00Q and 00L) or as interviewee 00B puts it: "*If we are going to make any headway, it is by going through the hearts and minds of the community*".

Detailed lessons learned from experiences in the implementation of on HEC management gathered from the expert interviews are summarized in the Discussion.

**Table 2.** Phase 1 results: HEC-management measures and tools grouped in six areas of HEC management operation, supported by quotations from expert interviews.

| HEC Management Strategy | HEC Management Tools | Example | Literature |
|---|---|---|---|
| Legel environment | *"We need that enabling environment to be established, [so that] everyone can work towards the policies, all working together." Statement by interviewee 00Q* | | |
| | International frameworks | Convention on Biological Diversity; Convention on Migratory Species | [94,95] |
| | Bi- and multi-lateral agreements | COMIFAC; SADAC | [96,97] |
| | National HWC strategies and action plans | National Policy on HWC Management, 2018–2027 Namibia; National HWC Management Strategy 2020–2024; Stratégie Nationale et Plan d'Actions de Gestion des Conflits Homme-Faune au Gabon | [98–101] |
| | Regional level guidelines | Problem Elephant Control guideline; Sustainable development guideline; Standard Operation Procedures (SOPs) | [102,103] |
| | Regional/local level by-laws | By-laws and their enforcement in bear-smart community program, Canada | [104] |
| Spatial management | *"Corridors and connectivity need to be secured, even if elephant populations are low." Statement by interviewee 00E* | | |
| | Spatial population planning | Corridor plans; Maintenance of contiguous populations; Consideration of infrastructure planning (roads, mining, fences, tourism, etc.) | [105,106] |
| | Land-use planning on landscape level, zonation | Elephant zones, peoples areas, overlap areas (e.g., natural resource use areas) | [107–109] |
| | Participatory land-use plans on micro (local) level | Participatory elephant corridors; Forestry frameworks | [110–112] |
| | Definition of agricultural practices for specific zones | Block agriculture instead of spreading and shifting cultivation; Improvement of agriculture; Soil protection; Conservation farming to achieve more yields on smaller space and decrease shifting; Production and protection of value-added crops; Cultivation of crops unattractive to elephants | [113–116] |
| | Definition of habitation areas | Development of permanent settlements; Improved and safe housing and storage | [117,118] |
| | Strategic water supply | Separating water sources for people and elephants; Creating safe water access for people | [7,119,120] |

**Table 2.** *Cont.*

| HEC Management Strategy | HEC Management Tools | Example | Literature |
|---|---|---|---|
| **Social strategies** | *"Only the community itself holds solutions for the future"* Statement by interviewee 00O | | |
| | Systematic stakeholder analysis | Stakeholder identification and mapping; Comparative network analysis and stakeholders' individual perception study | [121,122] |
| | Strategic and meaningful stakeholder engagement | Regional and local stakeholder fora and platforms | [47,121,122] |
| | Suitable institutional frameworks | Building strong community representation; Capacity building for community-based institutions | [123–126] |
| | Communication strategies | Choosing appropriate communication tools and language | [37,127,128] |
| | Community outreach | Mass awareness (drama plays, radio programs, sports events); Dialogue (information during community gathering, platforms); Training (for specific target groups); Exchange programs (for local leaders, decision makers, role models) | [129–133] |
| | Formal and informal school programs | Education material for various subjects; Teachers training; School clubs | [134–138] |
| **Technical strategies** | *"In many cases programs have trained elephants to become effective at avoiding or dealing with interventions. This is the worst thing we can do."* Statement by interviewee 00U | | |
| | Permanent exclusionary devices | Fences; Trenches; Barriers around/along protected areas, farms and/or habitations; Barriers along roads to avoid collisions with vehicles; Barriers to prevent disease transmission from wildlife to livestock | [34,139,140] |
| | Mobile exclusionary devices | Mobile electric fences for farms and habitations | [34] |
| | Deterrent devices | Acoustic (siren, sounds); Visual (torches, fire); Olfactory (chili smoke, organic smelly repellent) | [7,141,142] |
| | Deterrent fences | Chili fence; Beehive fence; Electric fence | [143–147] |
| | Combined deterrents | Chili bomber; Strategic community-based guarding; Unmanned aerial vehicles; HEC Rapid response | [148–151] |
| | Decreasing attractiveness | Change in agriculture; Production of unattractive (HWC smart) crops; Decreasing availability of food in villages | [113,152,153] |
| | Securing water points | Elephant-safe tanks and wells | [154] |
| | Early warning systems | Satellite tracking and geofences; Infrasound detection; Hotlines and farmers alerts | [155–158] |
| | Removal of problematic elephants | Translocation; Problem elephant control | [159,160] |

**Table 2.** *Cont.*

| HEC Management Strategy | HEC Management Tools | Example | Literature |
|---|---|---|---|
| Financial strategies | *Creating sustainable income from living with elephants has softened the friction in several communities. [This] is our trump card."* Statement by interviewee 00T | | |
| | Compensation of losses | Governmental schemes; Private initiatives | [161–163] |
| | Insurance schemes | Community-based insurance; Private insurance | [164,165] |
| | Benefits through living with wildlife | Revenue sharing schemes; Conservation performance payments; Elephant-based business | [166–169] |
| Monitoring | *"A standardized monitoring of HEC is key to informed decision making",* Statement by interviewee 00J | | |
| | Elephant movement/dispersal monitoring | GPS/satellite telemetry | [170–173] |
| | HEC assessment | Community-based monitoring; Agency-based monitoring | [139,174–176] |
| | HEC management monitoring | Community-based management action monitoring; Agency-based monitoring and evaluation; Data management and analysis | [69,177] |

*3.2. HEC Management Framework Adaptation and Validation*

In the adaptation and validation phase, the HEC management framework was field tested in specific project environments and contexts. Situation analysis and stakeholder discussions about tailoring the framework to local conditions revealed several insights. Key stakeholders from governmental and non-governmental conservation agencies, rural development, forestry, agriculture sectors, and local administrations supported the understanding that HEC will not be solved by solely implementing technical methods to stop wildlife entering farms and habitations. While the intensive stakeholder validation processes emphasized the requirement of all six strategies to be involved in an integrated way, some further requirements were identified: (1) Multi-species HWC management - due to ecological interlinkages and multiple-species damage scenarios; (2) Structure and fit of institutions—suitable institutional arrangements to implement the HEC/HWC management framework; (3) Processes and principles—comprehensive processes need to be carefully designed and community governance strengthened; (4) Placing social strategies in the center—the core of a holistic HWC management framework is the social strategies that bring stakeholders together and ensure communication, understanding, and enable engagement. Results from phase 1 and phase 2 were integrated into a single integrated HEC/HWC-management framework illustrated in Figure 3.

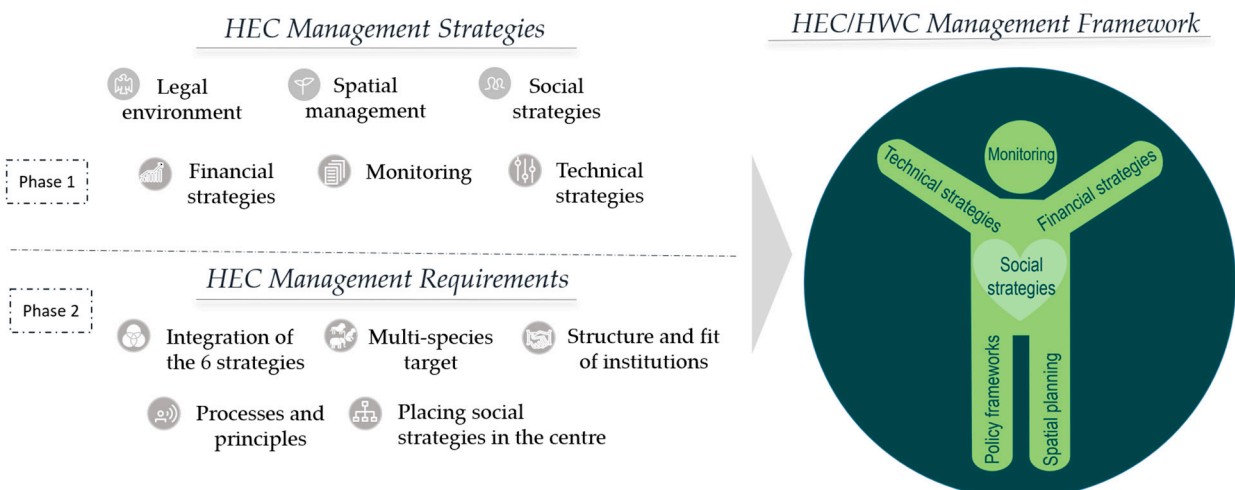

**Figure 3.** Illustration of the integration of phase 1 and phase 2 results into the HEC/HWC-management framework.

## 4. Discussion

The results obtained in phase 1 and phase 2 of this study led to the development of a HEC/HWC-management framework, which encompasses six overarching HEC management strategies and four further HEC/HWC-management requirements.

*4.1. Six HEC-Management Strategies*

4.1.1. Legal Environment

Various international framework agreements and national policies affect the management of HEC and the development of coexistence strategies. HEC management needs to be designed and implemented against the background of this legal environment. This is particularly important, as HEC management is a cross-sectoral theme, touching not only upon wildlife conservation, but upon agricultural development, forestry, water, veterinary issues, safety, etc. To foster the mainstreaming of HEC management and coexistence strategies into multiple sectors, it needs to be understood how current legal frameworks directly or indirectly affect HEC and its management. Furthermore, legal frameworks require harmonization to avoid unintended negative side-effects (e.g., between development

and conservation frameworks), and policy gaps need to be filled in (e.g., by well targeted regional bylaws) to effectively enable integrated HEC/HWC management and coexistence strategies [24,104,178,179].

An important international document for elephant conservation and HEC management is the African Elephant Action Plan (AEAP). The AEAP is fully owned and managed by the African elephant-range states and outlines the actions that must be taken in order to effectively conserve elephants in Africa across their range, including the reduction of HEC as the third of eight objectives [180]. Furthermore, HEC management is taken into account in a diverse range of international (e.g., CBD, CMS) and multilateral (SADC, COMIFAC) agreements [94,96,97]. Furthermore, the AEAP has been customized into National Elephant Action Plans by a range of countries (e.g., Malawi and Mozambique), and some included HEC management within conservation planning or as a separate subject. Overall, political directives on national levels (e.g., National HEC/HWC management action plans) are crucial to support the successful implementation of integrated long-term coexistence strategies for people and elephants. National strategies guide HEC-management processes on a regional level and may define specific processes through species and subject specific guidelines.

4.1.2. Spatial Management for Human–Elephant Coexistence

When trying to find an answer to the question of whether elephants and people can coexist, the spatial scale has to be considered. Currently, people in 37 African countries are sharing their land and resources with elephants. In some areas, fragmented populations of elephants remain in isolated national parks, fully separated from human activities. In other areas, multiple-use zones provide habitat for elephants and space for people in a land-use mosaic. As elephants consume staple crops, compete for water resources, and damage homes to access food, a separation of people and farming activities at the fine spatial scales is necessary to avoid damage [117]. If solutions for a separation are found at these fine spatial scales, a coexistence at the large spatial scale may become possible.

The basic requirement for the separation at fine spatial scales is to understand the needs of elephants and the needs of people [5]. Landscape connectivity is crucial for thriving elephant populations in many African landscapes [181,182]. A strong emphasis has been put on understanding and designing the connectivity of elephant habitats to ensure movement and genetic exchange [183]. Furthermore, maintaining the connectivity of habitats has the potential to reduce the intensity of HEC. However, in case corridors are not properly protected and people living in its vicinity are not bound into economic strategies, education, and HEC-management programs, there is a high risk that such areas become sinks for elephants [184].

Thus, the spatial organization of a landscape through land-use planning and zonation on micro and macro levels are key tools. Land requirements need to be assessed, development goals need to be agreed upon, and space for income generation and development needs to be defined [109,110]. To enable planning based on understanding and acceptance, a fully participatory process is required. Including transparent communication and equitable participation of local stakeholders is indispensable for land-use planning, also to prevent mistrust and unjust appropriation of land (green grabbing) [185,186]. This process may take time, as most communities do not readily know what they exactly need now or in the future, and the community itself may not have homogenous needs. As different people may have different desires, attitudes, and identities [187], successful land-use planning requires professionally mediated processes on the local level and comprehensive discussions on the use of farming practices to achieve resilient consent. Thus, landscape approaches to conservation acknowledge the need for connectivity between actors to increase information sharing and trust building and to address shared problems [188]. Finally, the potential of a planning document will only unfold if all stakeholders of the region agree on it and contribute to its implementation.

### 4.1.3. Social Strategies for HEC Management

Social strategies for HEC management include participatory and inclusive stakeholder engagement, community outreach, and education.

HEC is defined as a conflict between people over elephant and land management, mostly about ownership and the use of natural resources. Such conflicts can only be resolved by the inclusion of all parties taking a role in resolving that very conflict. Thus, the participation of all interest groups plays a vital role in HEC management. Furthermore, HEC management has to be sustainable in the long-term, and, therefore, it should be ideally administered by the local community itself. However, African elephants, with their high protection status, fall under the protection of the government. Therefore, communities and governmental authorities need to work together transparently. Political measures, good governance, and the creation of trust and reliability can only be achieved through inclusion and participation. Non-transparent governance (even within community-based programs) ultimately leads to negative attitudes towards the wildlife authority and its representatives [32,189].

Building up trustworthy work relationships based on mutual respect and understanding between parties directly involved in HEC issues is essential. This can be achieved by involving communities and their representatives, district administration, tourism companies, agricultural institutions, forestry, conservation NGOs, and other stakeholders into an open dialogue with shared information. In general, meaningful local participation with clearly defined roles and strong community ownership of the process will lead to higher acceptance of and tolerance towards conservation work [190–192]. It is highly recommended that the facilitation of any platform or forum is carefully designed by a professional facilitator. Power imbalances may lead to the dominance of some individuals or personal agendas, at the expense of others, who are not heard, making them feel marginalized, and potentially leading to or exacerbating conflict [121].

In terms of HEC management, NGOs often see themselves as a bridge between the community and governmental authority. Limitations of such concepts are seen in short-term funding of NGOs, particularly by international organizations. Furthermore, such concepts have their limits, particularly when transparency is lacking. Thorough planning and adequate capacity building are required for conservation NGOs to support law enforcement and engage in HEC management and community work. This demands transparent communication amongst stakeholders to ensure that the abuse of power is prohibited and that human rights are safeguarded at all times.

Additionally, specifically designed educational activities and the raising of awareness about the importance of elephant conservation are regarded as crucial for the long-term coexistence between elephants and people. In addition to the formal and informal education if pupils, this involves capacity building of various groups, such as community and agency-based rapid response teams, farm guards, community representatives, communication personnel. However, only a few programs seem to work on this topic in a strategic way. If HEC programs are to achieve long-term success for the safe coexistence of people and elephants, educational and outreach programs need to be reviewed and objectively evaluated.

### 4.1.4. Technical Strategies for HEC Management

A diverse arsenal of technical measures to prevent and decrease crop damage by elephants has been developed over the past decades. Most of these technical strategies are proposed to be used by farmers or community members across large rural landscapes, whereas affordability, practicality, and resistance to habituation are seen as the most important criteria [193]. As elephants have a strong capacity to learn, they can easily adapt to mitigation strategies and habituate to deterrents. Moreover, the multiple causes for their behavior need to be explored, understood, and considered [5,194]. Furthermore, the implementation of technical strategies may differ with regard to the ecological and cultural contexts. For this reason, every technical strategy has its limitations, and the circumstances

under which they work or not have to be taken into consideration. It is crucial to understand that one technical tool to solve HEC does not exist. A combination of short-term and long-term measures needs to be designed into a well-thought-out strategy, which allows adjustment and flexibility. Ad hoc activities to soothe high levels of damage have to be distinguished from strategies to decrease damage in the long-term.

Exclusionary methods, such as electric fences or trenches, separate elephants and people. Placed around a protected area, they are meant to keep elephants in and out of human dominated landscapes. At the same time, fences keep people and their livestock out of protected areas. Moreover, fences decrease the movement of wildlife and result in high costs for population management. Furthermore, maintenance costs and labor are high, so many fencing projects have failed, as budgets did not take maintenance into consideration. Mobile electric fences can be set up around habitations or farming areas, where the monitoring of maintenance and usage is crucial. If not well maintained, fences can easily be broken down by elephants by pushing poles, snapping electric lines with their tusks, or felling trees over the fences.

When exposed to acoustic, visual, or olfactory deterrents, elephants should develop fear and respond with flight. Ideally, deterred elephants should learn to avoid areas protected with deterrents, resulting in a long-term protective effect; however, habituation is a serious concern. If an elephant is continuously confronted with a deterrent which is unpleasant but not life threatening, and at the same time manages to gain a positive response to its behavior through feeding, the feeding success will outweigh the deterrent effect of the measure.

Strategic community-based guarding and rapid responses involve multiple deterrent measures and improve the traditional guarding practice of local farmers. Therefore, a common protection line is defined, to which all guarding efforts are shifted. Thus, the protection of a whole farming block or area can be achieved through cohesive and strategic efforts. Multiple deterrent methods are employed in strategic guarding and rapid responses, such as the chili bomber, a device combining acoustic, physical, and olfactory deterrence. It is a simple device to shoot elephants with ping-pong balls filled with a chili-oil extract. The ping-pong is fired with strong force, so that it breaks when hitting the skin. The chili bombers can be produced locally and used by trained community members.

Unmanned aerial vehicles (UAVs) can be used as a modern combined deterrent to chase away elephants from the air [149]. Operation is conducted by skilled and trained UAV pilots, for example by members of rapid HEC response teams. Such trained experts on driving away elephants should be employed by governmental or non-governmental organizations.

Deterrent fences are low-cost physical barriers, which are supplemented with some deterrent measures to make them more effective against elephants (e.g., chili fences or metal-strip fences). The beehive fence consists of a fence onto which beehives are attached, and the hives are naturally populated by wild honeybees. Once elephants try to enter through the fence, bees will start buzzing in the hives, which deters the elephants. The deterrent effects depend on the occupation rates of hives with bees, which, coupled with market-based strategies, have the potential for income generation [144].

All measures described above are measures to protect highly attractive crops or other food sources from elephants. As feeding preference of elephants on staple crops or other readily available food on farmland and villages will persist, such measures will always be cost- and labor-intensive. Decreasing the attractiveness of farms and villages to elephants, is another strategy to decrease the presence of elephants in human-dominated areas. This includes the farming and marketing of crops which contain essential oils or other antifeedants and, thus, are avoided by elephants [113,195]. Furthermore, the attractiveness of villages to elephants can be reduced by storing edible products in elephant-safe containers, dumping garbage in safe pits, and securing water points.

Reliable early warning systems, to detect elephants in a specific area and to warn farmers of their presence, are still under development [156,157]. Organizations and institutions are experimenting with satellite tracking and geofencing, infrasound detection or other

alarm systems. Great achievements in real-time tracking of elephants have been supported by tech-based early warnings; however, the response to alarms still needs to be performed manually and is risky and labor- and cost-intensive.

The so-called problem elephants, which are habituated to humans' presence, have learned where to find highly nutritious crops and how to undo crop-protection measures, may have to be removed from an area as a last resort. The translocation of such individuals is widely advocated by animal rights groups. However, this measure is highly cost-intensive and translocated elephants often return to their original territory. Problem elephant control should be seen as the last available means to deal with elephants displaying very problematic behavior [159]. However, it needs to be taken into consideration that when taking out one strongly habituated animal, it will most probably result in a new elephant taking the niche.

### 4.1.5. Financial Strategies for HEC Management

Losses caused by elephants to the farming community include direct damage caused by trampling, feeding on crops, or the destruction of houses, stores, or other infrastructure. Indirect costs are associated with high labor investments for guarding or the maintenance of crop protection measures and detrimental effects through guarding at night, exposure to diseases, absence from school for guarding, etc. [30,196].

Although conservation organizations, sport hunters, and tourism may be the largest employers in many HEC areas, limitations are observed in income generation through the presence of wildlife [169,197,198]. Generated funds are generally used for community development projects and not to offset losses on an individual level. The call by farmers to compensate for or at least offset losses is comprehensible if no benefits achieved from the presence of elephants. Financial strategies to mitigate HEC need to consider offsetting costs and increasing benefits of living with elephants.

Financial compensation of wildlife damage is discussed controversially, and functional governmental compensation schemes are rarely found on the African continent. Slow processes, difficult accessibility, and low compensation amounts are points of criticism in areas where compensation schemes do exist [164]. In case governmental compensation schemes are inexistent or fail, communal or private initiatives can give relief to farmers experiencing severe damage. Payments are tied to damage incidents and are funded at least partially through premium/membership payments. While increasing numbers of innovative schemes are being launched with a focus on providing enhanced protection for climate-related weather impacts to smallholders across the African continent, insurance schemes for HEC-related incidents have been limited in scope [164]. To be successful, the implementation of insurance schemes must be bound into holistic and integrated HEC-management schemes, be closely linked with preventive measures, take into consideration the social and ecological context, be site specific, and build upon stakeholder trust and effective monitoring of success. The main challenges of such schemes are related to the timely and accurate verification of damage, clear rules and guidelines, prompt and fair payment, and sufficient sustainable funds. Community-based insurance schemes can be based on revolving funds filled by income generated through the presence of wildlife. However, to design any sustainable and equitable compensation scheme, calculations need to be based on accurate and realistic damage data.

Furthermore, revenue-sharing concepts were developed in Pas, based on the assumption that the presence of wildlife can create enough income for a community to bear the costs of coexistence. A common concept is that income generated by Pas (e.g., through entrance fees) is used to contribute to community-management structures. Although these concepts lead to a positive direction, shares for communities often are insufficient and, in the case of governance constraints, can easily be misused [168,199,200]. Although community members may appreciate the community-development activities financed by revenue-sharing schemes, their individual losses are not met, which leads to the feeling of

inequity. Thus, economic and marked-based strategies benefitting individuals can play an important role in increasing individual resilience.

Conservation performance payments are gaining momentum on the African continent. Thereby, communities are rewarded for compliance with conservation and for tolerating wildlife in their neighborhoods [166,201–203]. Performance payments are paid to the community according to jointly developed criteria. Coexistence performance increases payment; negative performance decreases payment. The strength of the performance payments is the direct and transparent link of conservation to a benefit. Coupling performance payments with offsetting losses by a community-led insurance scheme might be a viable option for community-based human-wildlife coexistence strategies.

### 4.1.6. Monitoring of HEC and Its Management

HEC monitoring informs evidence-based decisions in HEC management and includes the collection of empirical HEC data, considering ecological, social, and management factors. Data on the frequency and magnitude of crop and property damage, number and background of peoples' and elephants' fatalities, as well as people's attitudes, define the character of the situation. Objective data collection is required, because subjective data cannot be related to the real extent. Integrating the monitoring of HEC-management measures fosters learning about the effectiveness of single and combined strategies. Furthermore, the spatial location of the damage is seen as an important feature for mapping areas of high damage and capturing changes over time. The combination of such HEC data with elephant movement and dispersal, as well as with poaching data may provide direction for future HEC-management applications and for the development and operation of sound and smart indicators for HEC management.

In the 1990s, a first approach was made to produce more comparable data through publishing a standardized format for the assessment of damage caused by elephants [204]. However, this effort was focusing on the damage caused to an individual farmer only and did not take into consideration the use of mitigation strategies or differentiate between damage caused by trampling and feeding. Data collection on individual farmers' levels created the bias of autocorrelation when multiple farmers' fields were damaged at the same time. As proposed by Naughton-Treves [176] and Gross, Lahkar, Subedi, Nyirenda, Lichtenfeld and Jakoby [175], damage events should be defined as damage by an individual or group of one wildlife species during one time period (e.g., one night) in a defined area, to decrease the likeliness of autocorrelation.

Obtaining a good coverage of HEC monitoring is labor- and cost-intensive. While upscaling on tech-based data collection within wildlife authorities, involving communities in HEC monitoring is another way to create sustainability in monitoring. Community-based monitoring data enables great coverage, first-hand data for all community-based actions and creates ownership over data, highlighting outcomes and performance [81,177,205].

### 4.2. Four Further HEC/HWC-Management Requirements
### 4.2.1. Multi-Species HWC Management

As damage caused by one wildlife species inevitably influences the perception towards damage caused by other species, a single-species approach for HEC management was regarded as insufficient by multiple stakeholder groups. In the adaptation and validation process, the HEC-management framework was therefore extended to a HWC-management framework. Understanding the causes and drivers of HWC in its holistic entirety requires consideration of the needs and reactions of all living beings involved. While human perspectives towards HWC may strongly vary between different groups of people, the wildlife perspective adds to the complexity but requires consideration if coexistence is the aim [206].

### 4.2.2. Structure and Fit of Institutions

Suitable institutional arrangements to implement the HEC/HWC-management framework were identified as a central requirement. In socio-ecological systems, such as coexistence landscapes, the requirements for such structures are complex, because institutional structures need to serve both, development, and conservation. In order to ensure robust community participation under the HEC/HWC framework, there is the need to establish strong community institutions. Each institution on each administrative level plays an important role and takes on specific duties and responsibilities. In both project areas, the HEC/HWC-management framework was adapted to the present community-based natural-resource-management structures. At the grassroots level, the Natural resource committees (NRC in Malawi and Comités des Gestão de Recursos Naturais (CGRN) in Mozambique), which are elected through a democratic process by their constituting communities, may represent one or more villages within an area and need to represent both genders. Each NRC has its own constitution, which sets out the detailed procedures of how an institution will manage its daily affairs. This institutional level was assigned to coordinate and monitor the community-based HEC/HWC-management activities, e.g., monitoring of damage, production of resource-use maps, community-based HEC/HWC response, strengthening community awareness, and definition of community processes and criteria (see below). NRC chairpersons form strong regional representations (Zones in Malawi, Localidade in Mozambique), which exchange regularly and have the responsibility to supervise NRC actions, create by-laws, and manage funds for HEC/HWC actions. This institutional level requires a strong capacity and administration, access to bank account, etc., and works in close collaboration with a coexistence management unit. The coexistence management unit is established to build a special team charged with general oversight over community-based HEC/HWC management and coexistence strategies, monitoring and financial resources, agency-based rapid HEC/HWC response, and professional wildlife management. The Coexistence Unit is composed of representatives from conservation, development, and local administration (e.g., district) as well as community representatives. In addition to community institutions, governmental agencies as well as CBOs play a vital role in the implementation of an integrated and holistic HEC/HWC management framework.

Once tailored to the specific socio-demographic and administrative situation, the HEC/HWC-management framework is fit for further longer-term actions towards coexistence, such as livelihood diversification, benefit sharing schemes, disaster management, community education, etc.

### 4.2.3. Processes and Principles

Communities in the border zone around VMWR and NNP are key stakeholders and active partners in HEC/HWC management. Coexistence strategies need to be based on strong community involvement or even to be driven by the community itself. The level of community engagement and ownership is influenced by the conservation management concept and the national/regional frameworks for community engagement in place. With a shift from hierarchical to alternative approaches seeking the involvement of the community actors and the private sector, suitable governance models need to be developed. Along with the structures, comprehensive processes need to be carefully designed and community governance strengthened. Good governance principles for protected areas [207] can also be applied for social–ecological systems [208] and for community representations in VMWR and NNP and NSR.

From the onset of locally adapting the HEC/HWC-management framework, the partners need to consider a common goal and set of objectives to work towards and clearly set out each of the partnership interests. This requires joint planning among the respective partners and regular planning meetings with the players. Written partnership agreements will ensure transparency related to roles and responsibilities. While each partner must focus on delivering its partnership objectives and targets, they need the requisite resources to support the implementation of set activities. Therefore, each partner

should allocate sufficient people, budget, and other resources to allow the attainment of the desired outputs. Sufficient review and monitoring processes need to be installed and institutionalized for troubleshooting, examining, and evaluating the performance of integrated HEC/HWC management.

### 4.2.4. Placing Social Strategies in the Centre

While all six strategies are required to address the full complexity of HEC/HWC and address its challenges on various levels, stakeholder discussions and validation workshops have emphasized the central importance of social strategies. HEC/HWC management is about sharing experiences, building relationships, and trust, and it has to be mainstreamed into all sectors of conservation and development. Furthermore, any holistic HEC/HWC-management framework needs to be based on a strong and supportive foundation of legal frameworks guiding its actions and spatial planning with an ecological and socio-economic overview and foresight. If the social strategies are the heart of the framework, the technical and financial strategies bear powerful tools to create safe spaces for people and wildlife and to balance the costs of living with wildlife and support fair shares of benefits. On top of this, a holistic and integrated HWCMS builds upon evidence-based data provided by continuous monitoring and evaluation. In short, an effective HWCMS decreases the negative impact of wildlife on peoples' safety and livelihoods, while simultaneously decreasing negative impacts on wildlife and natural resources and maintaining or strengthening positive impacts.

### 5. Conclusions

Assuming the recent poaching crisis was solved in some parts on the African continent, HEC is likely to rise. Governmental and non-governmental institutions, therefore, need to prepare, before farmers unnecessarily suffer big losses. Responsible institutions, however, are facing real constraints in terms of HEC-mitigation strategies. In particular, they need to build up structure, capacity, and skills.

This HEC/HWC-management framework offers a validated and comprehensive framework for a holistic and integrated approach to human–elephant coexistence. Depending on the social, ecological, and economic context, its components and strategies can be tailored to meet the specific requirements. However, for the HEC/HWC-management framework to succeed, concerted efforts, intra-agency collaboration, and coordination are required (e.g., the conservation agency has the skills; the community has the workforce). Focusing only on one aspect, e.g., technical measures, will not bring the desired outcomes and will not create sustainability. Furthermore, integrated monitoring needs to provide constant feedback on progress and changing factors, to adapt HEC/HWC management as required.

While development shall be fostered, negative side-effects to the ecosystem need to be prevented and mitigated, while simultaneously conservation compliance of communities and individuals is enhanced. Ideally, such a system enables sustainable development on the local and regional levels and supports communities to take over a stewardship role for the environment, including its wildlife. For this, revenues generated through the presence of wildlife and ecosystem services will serve the community to firstly decrease the negative impact of living with wildlife and secondly enable sustainable development. In coexistence landscapes, damage will never be completely eradicated, thus, the creation of tangible benefits plays a major role in outweighing tangible and intangible costs. When starting the HEC/HWC-management framework, achieving tangible benefits will be crucial, particularly as short-term goals. Eventually, with diversified livelihoods, increasing income, higher independence, and resilience on household levels, benefits may further support community projects. Focusing on opportunities and benefits obtained from living with wildlife will further foster remaining connections and cultural identities related to nature.

Organizations and institutions working on HEC need to take into consideration the full complexity of the subject. This requires interdisciplinary work for which skilled labor and experts are needed. The careful development of adequate structures, capacities, and

processes requires intensive time investment. Particularly, the integration of community-interest groups on all levels calls for intensive participatory approaches, which can be time consuming. However, to be able to achieve high level of approval, which is the basis for long-lasting solutions, these investments have to be made. If people and elephants shall still coexist in the year 2050 on the African continent, such processes need to take a central role in elephant conservation as well as in rural development programs.

This framework has been developed with a focus on HEC in areas where elephants cause a major concern to local communities. However, conflicts were not limited to elephants, so a multi-species approach was embraced. Even though the framework was not explicitly tested for other species, we believe that it applies to other geographical landscapes (e.g., South and Southeast Asia) and other species (e.g., Asian elephants and other large herbivores and carnivores). However, explicit validation on site would still be required to create evidence for further broadening the scope of this framework.

**Supplementary Materials:** The following are available online at https://www.mdpi.com/article/10.3390/d14070525/s1: Interview guide phase 1.

**Author Contributions:** E.M.G. wrote the first draft, S.L. and J.G.P. contributed to the conception of the study and contributed revisions. S.B. and T.S. collected data for Phase 2 and contributed revisions together with B.K. All authors have read and agreed to the published version of the manuscript.

**Funding:** Phase 1 of this research was part of a study commissioned by the "Partnership against Poaching and Illegal Wildlife Trade in Africa and Asia", a project implemented by GIZ on behalf of the German Federal Ministry for Economic Cooperation and Development (BMZ) and the Federal Ministry for the Environment, Nature Conservation and Nuclear Safety, and Consumer Protection (BMUV). Phase 2 of this research was part of a study in Mozambique and in Malawi. The study in Mozambique was commissioned by the Wildlife Conservation Trust Mozambique and funded through the "Partnership against Poaching and Illegal Wildlife Trade in Africa and Asia" (predecessor of phase 2 "Partnership against Wildlife Crime in Africa and Asia"), implemented by GIZ on behalf of BMZ and BMUV. The study in Malawi was commissioned by the German Development Bank (KfW) and Transfrontier Management Unit (TMU) of Peace Parks Foundation (PPF) with funding by the German Federal Ministry for Economic Cooperation and Development (BMZ).

**Institutional Review Board Statement:** The authors confirm this material is the authors' own original work, which has not been previously published elsewhere. The paper reflects the authors' own research and analysis in a truthful and complete manner. The paper properly credits the meaningful contributions of co-authors and co-researchers. The results are appropriately placed in the context of prior and existing research. The authors assert that all procedures contributing to this work comply with the applicable ethical standards of the relevant national and institutional committees on human experimentation and with the Helsinki Declaration of 1975. Informed consent was obtained from all individual participants involved in the study.

**Data Availability Statement:** Not applicable.

**Acknowledgments:** This research study was a collaborative effort of multiple institutions and people. We are highly grateful to all institutions that significantly contributed to the development and publication of this work, particularly the Department of National Parks and Wildlife (DNPW)/Malawi, Administração Nacional das Áreas de Conservação (ANAC)/Mozambique, Wildlife Conservation Society (WCS)/Mozambique. This work would not have been possible without the cooperation and support of multiple international and national experts, NGOs, and teams of local interviewers, who collected data and supported workshops in NSR, VMWR, and NNP, such as Niassa operators/concessions (Mariri, Luwire, Metapiri, Chuilexi) that contributed for data collection and MOM's Monitoring-oriented Management System. We thank all interviewees, experts, community groups, their traditional leaders, district councils, and stakeholder representatives for allocating time to this study and for sharing their knowledge and experiences. We thank the AfESG for sharing the elephant range layers for Figure 1 with us. We further acknowledge three anonymous reviewers for the positive feedback and the constructive and straightforward comments helping us to increase the quality of this paper.

**Conflicts of Interest:** The authors declare no conflict of interest.

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
