# Peer review of "Exploring Routes to Coexistence: Developing and Testing a Human–Elephant Conflict-Management Framework for African Elephant-Range Countries"

_diversity, doi:10.3390/d14070525_

Round 1

Reviewer 1 Report

Human elephant conflict is a complex and dynamic issue of high relevancy for the African continent – it further serves as a classic example of human wildlife conflict with global implications. The paper by Gross and others contains a very comprehensive study dealing with data (quality expert interviews) collected to grasp and better understand the dimensions behind HEC in 12 African countries. Although a bit lengthy, I think the paper is a useful contribution to the topic that deserves being published in the journal Diversity given some improvements. I think that the quality of the paper might be increased quickly and with reasonable effort. Overall, I have minor comments and suggestions but no major concerns whatsoever. Two issues appear important and should be addressed:

To me, the most important aspect is the issue of structure. The paper contains a very extensive results chapter, literature references are cited within this results chapter, there is no discussion chapter. I understand that in social sciences – obviously this contribution falls into this category – sometimes results and discussion are merged into one chapter – this is not common for life sciences, however. I think that a separation into results and discussion would help to better understand and tell apart: what is the contribution of the authors – the originally collected data – and what is interpretation and discussion of the results. This should, however, not lead to the manuscript becoming even longer. In contrary, the authors might use a changed structure (separation and differentiation between results and discussion) to be more focused and on point…

What also makes it difficult to understand the main findings is the fact that there are almost no figures or illustrations that summarize or illustrate the findings – ok, there is this very simple figure 3 - but this could be improved in order to show how it was linked with the findings of this study. Figure 1 is also very simple and plain but -ok, it might serve for the explanation of the methodological or experimental design. Figure 2 – the map – is by far the most appealing graphic item of the paper but: each map should be associated with a scale bar and a north arrow, moreover, the map is obviously an exact copy of a map from an internet source – its labelled in German thought he paper is written in English – the more I dig into it, the more I realize that also this map needs to be adjusted and better labelled…

The paper cites an impressive amount of references but I think that the quality could be enhanced, if additional ideas are incorporated: One is the understanding of conflict – the authors might also consider references suggesting to apply a more ecocentric view on HWC and also take a look from the perspective of wildlife and not only from the perspective of humans. Suitable articles have been published on this, e.g. recently in the journal Diversity.

When trying to reconstruct the population development of elephants in southern Africa, it may be useful to adopt a more broad approach and regard historical data but not only refer to the so-called “ivory crisis” (second page, first line), I strongly suggest references with an historic reconstruction of elephant range, such articles exist e.g. in terms of Namibia, where the historic distribution rangse of several large wildlife specvies, including elephants, were reconstructed based on hitoric data – very useful info that should be considered when reflecting about HEC in southern Africa.

in some countries, like Mozambique, civil war also played an important role besides poaching: you may have a look at studies dealing with the recolonization of elephants (amongst other large herbivores) of Limpopo Park, Mozambique – suitable references exist and were also published in the very same outlet than the MS in hand-

What would be very interesting in view of the conclusions chapter is the question, in how far are the results transferable / applicable to other regions, e.g. elephants in Asia or other megaherbivores in various regions?

I would appreciate to see an updated version of the paper and I wish the authors good luck.

Author Response

Human elephant conflict is a complex and dynamic issue of high relevancy for the African continent – it further serves as a classic example of human wildlife conflict with global implications. The paper by Gross and others contains a very comprehensive study dealing with data (quality expert interviews) collected to grasp and better understand the dimensions behind HEC in 12 African countries. Although a bit lengthy, I think the paper is a useful contribution to the topic that deserves being published in the journal Diversity given some improvements. I think that the quality of the paper might be increased quickly and with reasonable effort. Overall, I have minor comments and suggestions but no major concerns whatsoever.

> Thank you so much!

Two issues appear important and should be addressed:

To me, the most important aspect is the issue of structure. The paper contains a very extensive results chapter, literature references are cited within this results chapter, there is no discussion chapter. I understand that in social sciences – obviously this contribution falls into this category – sometimes results and discussion are merged into one chapter – this is not common for life sciences, however. I think that a separation into results and discussion would help to better understand and tell apart: what is the contribution of the authors – the originally collected data – and what is interpretation and discussion of the results. This should, however, not lead to the manuscript becoming even longer. In contrary, the authors might use a changed structure (separation and differentiation between results and discussion) to be more focused and on point…

> We fully understand your concerns regarding structure, as this was an intensive internal discussion among the co-authors. Our challenge is that we had worked in consecutive steps/phases and used the results of one step for the next. Results from the literature review were used for structuring the qualitative interviews and the results from the qualitative interviews were used for the validation/testing… To increase the understanding of what was exact outcomes/results of the steps/phases and what is interpretations, we have now separated a short result section from the discussion. As per request of Reviewer 2 we have added some quotations and references of the expert interviews into the result section. Everything captured in results is pure results. The discussion, however, may still cover quite some results and discussion parts, but taring these apart would decrease readability. In fact, results from qualitative expert interviews may appear more as discussion parts than results. We strongly hope that the separation of results and discussion increases comprehensiveness.

What also makes it difficult to understand the main findings is the fact that there are almost no figures or illustrations that summarize or illustrate the findings – ok, there is this very simple figure 3 - but this could be improved in order to show how it was linked with the findings of this study. Figure 1 is also very simple and plain but -ok, it might serve for the explanation of the methodological or experimental design. Figure 2 – the map – is by far the most appealing graphic item of the paper but: each map should be associated with a scale bar and a north arrow, moreover, the map is obviously an exact copy of a map from an internet source – its labelled in German thought he paper is written in English – the more I dig into it, the more I realize that also this map needs to be adjusted and better labelled…

> Thanks for the remark. We have added content to Figure 3, which we believe has strongly added value to this figure and increases understanding on how results from Phase 1 and Phase 2 were integrated into the framework.  

> We have also revised Figure 2, the map. We have obtained approval for using the elephant range distribution map of the AfESG database, which we have used as background layers to illustrate the location of interview sites (phase 1) and study sites (phase 2).

The paper cites an impressive amount of references but I think that the quality could be enhanced, if additional ideas are incorporated: One is the understanding of conflict – the authors might also consider references suggesting to apply a more ecocentric view on HWC and also take a look from the perspective of wildlife and not only from the perspective of humans. Suitable articles have been published on this, e.g. recently in the journal Diversity.

> Thanks for pointing at this. We have included a short discourse on considering human and wildlife perspectives for understating HWC in the discussion “Understanding causes and drivers of HWC in its holistic entirety requires considering the needs and reactions of all living beings involved. While human perspectives towards HWC may strongly vary between different groups of people, the wildlife perspective adds to the complexity, but requires consideration if coexistence is the aim.” and used and used: Göttert, T.; Starik, N. Human–Wildlife Conflicts across Landscapes—General Applicability vs. Case Specificity. Diversity 2022, 14, 380. https://doi.org/10.3390/d14050380 as new reference

When trying to reconstruct the population development of elephants in southern Africa, it may be useful to adopt a more broad approach and regard historical data but not only refer to the so-called “ivory crisis” (second page, first line), I strongly suggest references with an historic reconstruction of elephant range, such articles exist e.g. in terms of Namibia, where the historic distribution rangse of several large wildlife specvies, including elephants, were reconstructed based on hitoric data – very useful info that should be considered when reflecting about HEC in southern Africa. In some countries, like Mozambique, civil war also played an important role besides poaching: you may have a look at studies dealing with the recolonization of elephants (amongst other large herbivores) of Limpopo Park, Mozambique – suitable references exist and were also published in the very same outlet than the MS in hand- 

> Thanks a lot for pointing at this limitation. We have added “and the end of decades of civil wars in various elephant range states” to this passage and used the references Daskin 2018, Lee&Graham 2009, Schlossberg, Chase, Griffin 2018, Craig, Gibson, Uiseb 2021. We have also added elephant populations were slowly recovering in some areas, to account for the fact that this is not the case in all parts of the elephant rage states.  

What would be very interesting in view of the conclusions chapter is the question, in how far are the results transferable / applicable to other regions, e.g. elephants in Asia or other megaherbivores in various regions?

> Thanks for this question. We have now addressed this in the last paragraph of the conclusion: “This framework has been developed with the focus on HEC in areas were elephants cause a major concern to local communities. However, conflicts were not limited to elephants, so that a multi-species approach was embraced. Even though the framework was not explicitly tested for other species, we believe that it applies to other geographical landscapes (e.g. South and Southeast Asia) and other species (e.g. Asian elephants and other large herbivores and carnivores). However, explicit validation on site would still be required to create evidence for further broadening the scope of this framework.”

 I would appreciate to see an updated version of the paper and I wish the authors good luck.

> Thank you!

Reviewer 2 Report

I think this is an interesting, clear, well written and important study! But what I am currently missing in both the conclusion and also a bit in the introduction is a clear statement of what novelty this framework brings to a field with already many conceptual papers come out recently? What additional value did your interviews have besides the literature review, could you not only have analysed the literature? I am sure there is important benefit to the interviews, but please highlight this for the reader. The topics appear to already have been decided before the interviews if I understand your supplementary information correctly?

In general, I think it is a very interesting summary of all strategies available and used but I think if I did not know about the interviews or validation from the method section, the results would mainly have appeared a narrative literature review. Would it perhaps be possible to give some quotes or specific insights form the different interviews? If participants would be numbered anonymously or given pseudonyms it would still pass ethical review.

Just to make it clear: I definitely see great value in this study and this framework, and you clearly did a very impressive work getting all these experts for interviews and even validate results, I just fear the paper is currently not selling as much as it could, and wonder if its value is clear enough to be attractive to be easily widely adopted by readers. Some key bullitpoints with take home messages, for example in the figure 3 could also make it more easy for readers to implement your results perhaps?

Author Response

I think this is an interesting, clear, well written and important study! But what I am currently missing in both the conclusion and also a bit in the introduction is a clear statement of what novelty this framework brings to a field with already many conceptual papers come out recently?

> Thanks for pointing on the need to strengthen the novelty of the framework. We have considered emphasizing this in the introduction and the conclusion:

> Edited last sentence of introduction: “The framework offers scientists and practitioners an overview of which strategies and processes need to be concurrently or step-wise integrated to develop a sustainable and impactful change towards coexistence of people and elephants.”

> Edited second paragraph of conclusions: “This HEC/HWC management framework offers a validated and comprehensive framework for a holistic and integrated approach to human-elephant coexistence. De-pending on the social, ecological and economic context its components and strategies can be tailored to meet the specific requirements.”

What additional value did your interviews have besides the literature review, could you not only have analysed the literature? I am sure there is important benefit to the interviews, but please highlight this for the reader. The topics appear to already have been decided before the interviews if I understand your supplementary information correctly?

> Thanks for pointing this out. While we are explaining in the results part “During the expert interviews, these six areas were specifically examined, and insights about requirements and best practices were collected.”, this might not be sufficient for the reader. We have thus further explained in the first results paragraph: “Specific focus was set on understanding the effects, challenges and limitations of methods used as well as the interplay between them and further requirements for coexistence of people and elephants in the respective areas. Methods that were identified as effective and valuable by the experts were clustered into 32 tools (Table 2).”

In general, I think it is a very interesting summary of all strategies available and used but I think if I did not know about the interviews or validation from the method section, the results would mainly have appeared a narrative literature review. Would it perhaps be possible to give some quotes or specific insights form the different interviews? If participants would be numbered anonymously or given pseudonyms it would still pass ethical review.

> Thanks for pointing this out. In fact we did collect a wealth of quotes. We have integrated one quote per HEC management strategy in the results in Table 2. Furthermore we have integrated this passage on further lessons learned with strong reference to the interviews and quotes in the results:

“Further lessons learned from the expert interviews is that HEC is seen as a symptom, not a cause (interviewee 00P and 00U). Habitat loss and the arising competition for land and resources as well as other economic, political and social factors are seen as causes and drivers for HEC. In general it is understood that the problem of HEC cannot be wiped out completely (interviewee 00I and 00T). There will always be some risk of crop and property damage and negative perceptions by individuals. However, the risk of HEC to arise must be reduced to a tolerable level (interviewee 00H).

HEC is a highly complex phenomenon with many levels involved (political, com-munity, social, family, financial, tradition, culture, ecology) (interviewee 00Q) and is strongly dependent on the context (interviewee 00L). Simple solutions therefore are not to be expected and learning about HEC needs to continue (interviewee 00L). Due to its complexity holistic approaches are needed, which understand and tackle the problem from all sides (interviewee 00Q) and strong community participation is generally seen as the only way forward (interviewee 00A, 00B, 00N, 00O, 00Q, and 00T). The involvement of local communities into HEC management in a strong participatory way seems crucial for achieving a peaceful coexistence of people and elephants (interviewee 00Q and 00L) or as interviewee 00B puts it: “If we are going to make any headway, it is by going through the hearts and minds of the community”.”

Just to make it clear: I definitely see great value in this study and this framework, and you clearly did a very impressive work getting all these experts for interviews and even validate results, I just fear the paper is currently not selling as much as it could, and wonder if its value is clear enough to be attractive to be easily widely adopted by readers. Some key bullitpoints with take home messages, for example in the figure 3 could also make it more easy for readers to implement your results perhaps?

> Thanks, we have added the key outputs of Phase 1 and Phase 2 (HEC Management strategies and HEC management requirements) into the Figure 3

Reviewer 3 Report

This is a valuable summary of which short- and long-term elements need to be integrated for successful human-wildlife coexistence and the minimisation of competition for resources. However, it comes across as a review paper, rather than what the title suggests, testing findings from conflict mitigation throughout Africa and applying to, and testing in, new scenarios.

I like the Phases that you clearly define in Figure 1, however, it is not clear to me what was tested/developed and carried through to Phase 2.  Whilst two study areas were identified and clearly described, it is not clear what findings were integrated and how a management framework was tested or implemented.

If you can integrate more of your findings from the study sites into the results then it may be clearer as to what findings from Phase 1 have been taken forward and thus how others can apply your findings.

Minor considerations:

L38 This South African reference does not seem to fit the broad statement

L49 'in' instead of 'into'

L163 this sentence needs to be reworded slightly as can be misinterpreted to mean experts in the languages of English and French, rather than HEC/HWC

Consider this reference in relation to Welfare costs.

Mayberry, A. L., Hovorka, A. J. & Evans, K. E. Well-Being Impacts of Human-Elephant Conflict in Khumaga, Botswana: Exploring Visible and Hidden Dimensions. Conservation and Society 15, 280–291 (2017).

Author Response

This is a valuable summary of which short- and long-term elements need to be integrated for successful human-wildlife coexistence and the minimisation of competition for resources. However, it comes across as a review paper, rather than what the title suggests, testing findings from conflict mitigation throughout Africa and applying to, and testing in, new scenarios.

I like the Phases that you clearly define in Figure 1, however, it is not clear to me what was tested/developed and carried through to Phase 2.  Whilst two study areas were identified and clearly described, it is not clear what findings were integrated and how a management framework was tested or implemented.

If you can integrate more of your findings from the study sites into the results then it may be clearer as to what findings from Phase 1 have been taken forward and thus how others can apply your findings.

> Thank you so much for pointing at this weakness of the paper. Reviewer 1 raised a similar concern regarding the structure. We have now re-structured the paper to better differentiate between results and discussion. Our challenge is that we had worked in consecutive steps/phases and used the results of one step for the next. Results from the literature review were used for structuring the qualitative interviews and the results from the qualitative interviews were used for the validation/testing… To increase the understanding of what was exact outcomes/results of the steps/phases and what is interpretations, we have now separated a short result section from the discussion. Everything captured in results is pure results. The discussion, however, may still cover quite some results and discussion parts, but taring these apart would decrease readability. In fact, results from qualitative expert interviews may appear more as discussion parts than results. We strongly hope that the separation of results and discussion increases comprehensiveness.

Minor considerations:

L38 This South African reference does not seem to fit the broad statement

> While Sholes&Mennells book on “Elephant Management” clearly focusses on the special South African situation regarding elephant management, the chapter we are referring to here gives a broad overview of human-elephant interaction on the African continent in general. However, as this is not a primary source, but part of a review, we have added two primary literature sources, with Paleoanthropology and Ethnoelephantology focus.

- Klein, R. G. 1987. Reconstructing How Early People Exploited Animals: Problems and Prospects.11-45.

- Lev, M. a., and R. Barkai. 2016. Elephants are people, people are elephants: Human–proboscideans similarities as a case for cross cultural animal humanization in recent and Paleolithic times. Quaternary International 406:239-245.

L49 'in' instead of 'into'

> Thanks, this has been edited

L163 this sentence needs to be reworded slightly as can be misinterpreted to mean experts in the languages of English and French, rather than HEC/HWC

> Thanks, the sentence has been edited

Consider this reference in relation to Welfare costs.

Mayberry, A. L., Hovorka, A. J. & Evans, K. E. Well-Being Impacts of Human-Elephant Conflict in Khumaga, Botswana: Exploring Visible and Hidden Dimensions. Conservation and Society 15, 280–291 (2017).

> Thanks for pointing at this important work. We have included citation in the introduction, referring to intangible costs of living with elephants and in the discussion under “Social strategies for HEC management”.

Round 2

Reviewer 1 Report

Since my comments were carefully addressed, my recommendation is to accept the paper.

Best wishes to the authors

Author Response

Dear Reviewer,

Thank you so much for your positive feedback. 

We are only remaining with some minor comments from Reviewer 3, which we will address just now. 

With kind regards, 

Eva Gross and co-authors

Reviewer 3 Report

For Authors

L71 – 72 This sentence needs rewording, I not sure what is meant about HEC actually is a conflict between people over elephants and over resources. Do you mean, HEC is a conflict between people AND elephants…. Nor do I understand ‘social and political aspects need to be taken more into focus of HEC analysis’…..

L107 resulting in short TERM outcomes

L108 consider changing to ‘ Recent holistic and integrated HWC managed approaches have been developed…’

L110 Broaden the spectrum of what?

L113 is it really due to a rise in animal populations, not a rise in human populations increasing pressure on resources?

L116 write out abbreviation in full

L118 delete the ‘and ’ between acknowledges & the need

L130 consider changing ‘we here draft a framework’ to ‘in this paper we draft a framework’

L147 what does GIZ stand for?

L148 What do BMZ and BMUV stand for ?

L166 audio or video recordings? In person or via conference call/phone? Take from L175 and include here

L172 pluralise population

L175 & 177Analyses not analysis

Figure 2. Interviewee location may be better as the interviews took place remotely

Table 1 What is the difference between Elephant Crisis Fund, Save the Elephants and Save the Elephants?

L202 delete ‘number of ’

L206 replace ‘about’ with circa or estimated

L230 At editors discretion but usually when an animal is referred to the scientific name is included after the common name in brackets on the first use of the word.

When talking about the study area, an idea of the number of people bordering the protected areas that are detailed would give some insight into the HEC situation. Also are their any numbers on HEC incidents, to again give some insight into the level of HEC

L281, include loss of human life, injury and welfare costs. The ‘human perception’ is not HEC.

L282, this sentence does not make sense… do you men this, ‘However, the risk of HEC to arise must be reduced to a tolerable level

L283, do you mean factors rather than levels?

L289 The following sentence does not make sense and also I do not think you need it ‘In the adaptation and validation process of the HEC management framework to concrete project settings the concept was put to practical test’.

L303 delete ‘and means’

L312 exchange of what? Knowledge?

Table 2:

·      Perhaps merge columns 2,3&4 so that quotes go across top of each section

·      “We need [an] that enabling environment to be established……

·      What does SOP mean?

·      Separateing water sources from for people and elephants, creating safe water access to people

·      What is meant by shifting’?

·      Deterrent fences, include electric fences as referred to in L451 Make it clear that Problem Animal Control means shooting the elephant

L324: I think you mean phase 1 and phase 2

L359 delete ‘general’

L362, what does ‘strongly’ separated mean?

L364 ‘are fond of’ replace with eat or utilise.

L365 ‘may have learned to search’ consider rephasing to ‘and can damage homes to access food

L384 what does ‘green grabbing’ mean?

L385 in the future

L389 do you mean use and of farming practices?

L398  ‘Such conflicts can only be resolved by the inclusion of all parties taking a role in that very conflict’. Do you mean a role in resolving the conflict?

L402 there is no longer an African elephant as a species. So either refer to the African savannah elephant or say ‘the African elephant with its their high’

L404 delete the second therefore.

L430-431 in table 2 you refer to the importance pof educatoing childrenm, but do not list them here

L452, it could be argued that ‘human land’ is natural elephant habitat, rather say keep them in protected areas.

L474. Change It is a simple device to shoot ping-pong balls filled with a chilli-oil extract against elephants’ to ‘It is a simple device to shoot elephants with ping-pong balls filled with a chilli-oil extract against.

L476. Consider replace ‘on’site’ with ‘locally

L486, do the bees not also leave the hive?

L488 change ‘bears’ to ‘has’

Author Response

Responses to reviewers – point by point – Manuscript diversity-1734347

Reviewer 3:

Thank you so much for your very much for your thorough review and constructive suggestions, which are very much appreciated! Please find our pint-by point responses below.

L71 – 72 This sentence needs rewording, I not sure what is meant about HEC actually is a conflict between people over elephants and over resources. Do you mean, HEC is a conflict between people AND elephants…. Nor do I understand ‘social and political aspects need to be taken more into focus of HEC analysis’…..

> The first part of the sentence has been tweaked to “As HEC actually is a conflict between people about management of elephants and access to resources”.

> The second part of the sentence has been changed to “…the political economy and social dimensions need to be taken more into the focus….”, which is more specifically to the point.

L107 resulting in short TERM outcomes

> Changed as proposed. Thanks!

L108 consider changing to ‘ Recent holistic and integrated HWC managed approaches have been developed…’

> Changed as proposed. Thanks!

L110 Broaden the spectrum of what?

> What we meant by this that the perspective was broadened towards the drivers of HWC and their interrelation – we have tweaked into “Widening the angle from one species to multiple species, which may be inter-related, further fostered system thinking

L113 is it really due to a rise in animal populations, not a rise in human populations increasing pressure on resources?

> This is how the ToC of the Safe systems approach is described. “…to decouple the rise in the number of tigers from the rise in the number of human deaths” https://medium.com/together-possible/people-of-together-possible-ashley-brooks-124e8e038eb8

L116 write out abbreviation in full

> We have done so and updated to “Human-Wildlife Conflict & Coexistence Specialist Group of the International Union for Conservation of Nature (IUCN)”, as it is not a task force anymore but a SSC specialist group

L118 delete the ‘and ’ between acknowledges & the need

> Done. Thanks.

L130 consider changing ‘we here draft a framework’ to ‘in this paper we draft a framework’

> Done. Thanks.

L147 what does GIZ stand for?

> Added: German Corporation for International Cooperation GmbH

L148 What do BMZ and BMUV stand for ?

> Added: German Federal Ministry for Economic Cooperation and Development (BMZ) and the Federal Ministry for the Environment, Nature Conservation and Nuclear Safety and Consumer Protection (BMUV),

L166 audio or video recordings? In person or via conference call/phone? Take from L175 and include here

> Took from line 166 and included in line 175: “The interviews took 50 to 180 minutes and were carried out online as single expert interviews and were recorded and hand transcribed for analyses”

L172 pluralise population

> Done. Thanks.

L175 & 177Analyses not analysis

> Done. Thanks.

Figure 2. Interviewee location may be better as the interviews took place remotely

> The caption sais “locations of expert interviewees of this study”. We would like to keep the legend as it is, because we thing “interviewee location” may sound confusing. As an interview is carried out between at least two people we think “Interviews location” is not wrong.

Table 1 What is the difference between Elephant Crisis Fund, Save the Elephants and Save the Elephants?

> Elephants Crisis Fund is a programme under Save the elephants. We have omitted “Elephants Crisis Fund”, to maintain consistency within the table.  

L202 delete ‘number of ’

> Done. Thanks.

L206 replace ‘about’ with circa or estimated

> Replaced “about” with “circa”

L230 At editors discretion but usually when an animal is referred to the scientific name is included after the common name in brackets on the first use of the word.

> Fine, we have added: Loxodonta africana (Line 40);

> Line 206-208: Panthera leo, P. pardus, Lycaon pictus, Hippotragus niger, Tragelaphus strepsiceros, Connochaetes taurinus, Equus quagga

> Line 233-236: Aepyceros melampus, Antilope redunca, T. scriptus, Syncerus caffer, Crocuta crocuta, Lupulella adusta

When talking about the study area, an idea of the number of people bordering the protected areas that are detailed would give some insight into the HEC situation. Also are their any numbers on HEC incidents, to again give some insight into the level of HEC

> Added to Lien 257/258: “The human population within the study area adjacent to VMWR and the south-western NNP is estimated at 70.000 [79].”

> Unfortunately, a standardized assessment of HWC incidents does not exist in Vwaza/Nyika and figures for crop damage in Niassa are unreliable. In both areas HWC is a hot topic, meaning that it has become highly political and number of damage incidents describes only a part of the conflict.

L281, include loss of human life, injury and welfare costs. The ‘human perception’ is not HEC.

> This is results and we have to respect what interviewees say. This interviewee did not mention human life, injury and welfare costs. He/She mentioned negative perceptions.

L282, this sentence does not make sense… do you men this, ‘However, the risk of HEC to arise must be reduced to a tolerable level

> I agree that this sentence (quote) is hard to understand. I have gone back to the original recordings are listened to the content of this quote. The interviewee refereed to risk of damage. So I have changed the content accordingly.

L283, do you mean factors rather than levels?

> What the interviewee meant is that risk is reduced to an extent which can be tolerated (e.g. because there is alternative or access income). A “tolerable level” to me seems appropriate.

L289 The following sentence does not make sense and also I do not think you need it ‘In the adaptation and validation process of the HEC management framework to concrete project settings the concept was put to practical test’.

> We would like to keep this introductory sentence and have tweaked it to make it clearer. “In the adaptation and validation phase the HEC management framework was field tested in specific project environments and contexts”

L303 delete ‘and means’

> Done.

L312 exchange of what? Knowledge?

 > “exchange” has been replaced by “communication, understanding” – the exchange between stakeholders can of course be exchange of knowledge and experiences, but also simply developing common ground through listening and understanding.“ “Communication and understanding” seems appropriate to us.

Table 2:

  • Perhaps merge columns 2,3&4 so that quotes go across top of each section

> Yes, the original table looks like that. It does not show up properly in the track-change mode

  • “We need [an] that enabling environment to be established……

> I agree that this would be correct written language, but this is an interview quote and I would not like to change the language further.

  • What does SOP mean?

> Added: Standard Operation Procedures

  • Separateing water sources from for people and elephants, creating safe water access to people

>Changed “from” to “for”

  • What is meant by shifting’?

> Added shifting “cultivation”

  • Deterrent fences, include electric fences as referred to in L451

> Added electric fence to “deterrent fence”

Make it clear that Problem Animal Control means shooting the elephant

> PEC is a defined term and is listed under “removal”. We believe this does not require further explanation.

L324: I think you mean phase 1 and phase 2

> Thanks, Added.

L359 delete ‘general’

> done

L362, what does ‘strongly’ separated mean?

> “strongly” changed to “fully”

L364 ‘are fond of’ replace with eat or utilise.

> changed to “consume”

L365 ‘may have learned to search’ consider rephasing to ‘and can damage homes to access food

> done

L384 what does ‘green grabbing’ mean?

> added “unjust appropriation of land” to explain green grabbing

Wikipedia: Green grabbing, also known as green colonialism, is the foreign appropriation of land and resources for environmental purposes,[1] resulting in a pattern of unjust development.[2] The purposes of green grabbing are varied; it can be done for ecotourism, conservation of biodiversity or ecosystem services, for carbon emission trading, or for biofuel production. It involves governments, NGOs, and corporations, often working in alliances. Green grabs can result in local residents' displacement from land where they live or make their livelihoods.

L385 in the future

> Thanks, Added.

L389 do you mean use and of farming practices?

> Yes, thanks, changed.

L398  ‘Such conflicts can only be resolved by the inclusion of all parties taking a role in that very conflict’. Do you mean a role in resolving the conflict?

> Yes, thanks, changed.

L402 there is no longer an African elephant as a species. So either refer to the African savannah elephant or say ‘the African elephant with its their high’

> Yes, thanks, changed.

L404 delete the second therefore.

> Yes, thanks, changed.

L430-431 in table 2 you refer to the importance pof educatoing childrenm, but do not list them here

> Added” Besides formal and informal education for pupils...”

L452, it could be argued that ‘human land’ is natural elephant habitat, rather say keep them in protected areas.

> done.

L474. Change It is a simple device to shoot ping-pong balls filled with a chilli-oil extract against elephants’ to ‘It is a simple device to shoot elephants with ping-pong balls filled with a chilli-oil extract against.

> Thanks, changed accordingly.

L476. Consider replace ‘on’site’ with ‘locally

>Thanks, changed.

L486, do the bees not also leave the hive?

> The buzz already deters elephants. Yes, bees can also leave the hive (during daytime) or if very strongly disturbed. But they do not have to leave the hive to deter the elephant. Suggest to keep as it is.

L488 change ‘bears’ to ‘has’

>Thanks, changed.
